# A Comparison of Haematological and Biochemical Profiles between Intrauterine Growth Restriction and Normal Piglets at 72 Hours Postpartum

**DOI:** 10.3390/ani13223540

**Published:** 2023-11-16

**Authors:** Lucía Ayala, Cristian Jesús Sánchez, Fuensanta Hernández, Josefa Madrid, Miguel José López, Silvia Martínez-Miró

**Affiliations:** Department of Animal Production, Faculty of Veterinary, International Excellence Campus for Higher Education and Research “Campus Mare Nostrum”, University of Murcia, 30100 Murcia, Spain; lucia.ayalag@um.es (L.A.); cristianjesus.sanchez@um.es (C.J.S.); nutri@um.es (F.H.); alimen@um.es (J.M.); mjlopeza@um.es (M.J.L.)

**Keywords:** IUGR, piglets, haematology, biochemical profile

## Abstract

**Simple Summary:**

In recent years, the pig industry has focused on genetic selection based on prolificacy. However, this has resulted in heterogeneous litters, more low-birth-weight piglets, and an increase in the incidence of piglets being born with intrauterine growth restriction (IUGR), decreasing the survival rate. Thus, the aim of this study was to advance the haematological and biochemical knowledge of IUGR piglets in the first days of life and evaluate their possible changes compared to normal piglets. At 72 h of life, we found differences in the parameters studied. IUGR piglets showed changes in red and white blood cells, reticulocytes indices, and some biochemical parameters, which indicated immaturity. The results provide information on the analytical profile of IUGR piglets and suggest that the divergence of the two groups of piglets is evident a few hours after birth.

**Abstract:**

Intrauterine growth restriction in piglets has been a problem in the pig industry due to genetic selection based on hyperprolificacy. This has led to an increase in the number of underweight piglets and a worsening of the survival rate. The goal of this study was to enhance the knowledge of differences between normal and IUGR piglets a few hours after birth in terms of haematological variables, biochemical parameters, and immunoglobulin levels. Two groups of 20 piglets each were assessed. The control group (N) was made up of piglets with weights greater than 1500 g, and the IUGR group consisted of piglets weighing 500–1000 g and with at least two IUGR features. Blood samples were collected 72 h after birth for analysis of the red and white blood cell parameters, reticulocyte indices, platelet indices, biochemical parameters, and immunoglobulin levels. Alterations in red blood cells and reticulocytes, a lower lymphocyte count, hyperinsulinemia, and high oxidative stress were observed in IUGR piglets (*p* < 0.05). In contrast, differences were not observed (*p* > 0.05) in the serum immunoglobulin level. It can be concluded that the haematological and biochemical differences in IUGR piglets with respect to normal-weight piglets are present at birth indicating possible alterations in immunity, metabolism, and redox status; therefore, IUGR piglets could be more vulnerable to illness and future disorders, such as metabolic syndrome.

## 1. Introduction

One of the most important economic factors in pig production is prolificacy. For that, the objective of most breeding programs has been increasing the number of piglets per litter, as well as the litter survival [1]. Using genetic selection for highly prolific sows, litter size in sows has dramatically increased in recent decades [2].

However, associated with the appearance of hyperprolific sows, the within-litter variability and the number of small piglets born has increased, and the piglets’ mean birth weight has decreased. In addition, an increase in the frequency of piglets born with intrauterine growth restriction (IUGR) has also been observed. Around 30% of newborn pigs in Denmark showed signs of IUGR [2,3].

Intrauterine growth restriction is a condition in which the foetus and its organs grow and develop slowly or poorly during gestation [1,3,4]. This may occur when there is reduced blood flow to foetal tissues, and so the nutrients and oxygen needed for the growth and development of foetal organs and tissues are not available [5]. These alterations in foetal growth patterns result in a brain-sparing effect as part of a foetal adaptation to placental insufficiency. Nutrients are partitioned preferentially to the brain versus other tissues to ensure proper development of the brain [6]. Due to this effect, IUGR piglets have a distinctive head shape with a steep dolphin-like forehead [4] and other external characteristics such as bulging eyes, wrinkles on either side of their mouths, and no single direction of hair growth [7]. Another aspect of IUGR piglets is a lower rectal temperature at birth than normal piglets. These piglets have a larger surface area to body mass ratio than normal piglets, resulting in greater heat loss, which leaves them susceptible to hypothermia. This is exacerbated by the fact that IUGR piglets are born with lower whole-blood glucose and glycogen storage levels than normal piglets; it is therefore essential that these piglets ingest colostrum as soon as possible in order to maintain their body temperature and increase their energy sources. However, the lack of energy at birth makes it difficult for them to compete for udder space and most of them will not have sufficient access to colostrum, resulting in a high mortality rate [3]. Thus, there is no doubt that the weight of a piglet at birth affects its chance to survive [1]. In addition, IUGR piglets may have difficulty progressing through the birth canal and may face a higher risk of asphyxia during farrowing. Asphyxia during delivery is detrimental to the piglets’ survival and ability to adapt to life outside the womb. Accordingly, lactate levels are higher in IUGR piglets at birth than in normal piglets, suggesting a lower oxidative capacity [1]. Therefore, piglets born with IUGR face greater challenges at birth.

IUGR piglets also evince changes in organ structure and function. Compared to their littermates, IUGR piglets have significantly smaller livers, intestines, and pancreases. This effect is proportional to the loss in body weight. In the intestinal tract, reduced villus height and crypt depth, and leaner mucosa and muscle layers, can be observed. IUGR piglets have a smaller passive absorption area than non-IUGRs, and their gastric and pancreatic secretions also seem to be lower. Therefore, in IUGR piglets luminal digestion and absorption of nutrients is not as effective as it is in non-IUGRs. Due to the delay in digestive mucosa maturation, food changes and bacteria can be more difficult to adapt [7]. Regarding proteomic analysis, IUGR piglets’ intestines showed a decrease in the concentration of proteins implicated in iron metabolism, such as transferrin, and other proteins like retinol binding protein 2 (RBP2). Lack of RBP2 reduces growth and immunity, and has an effect on adipocyte differentiation, lipolysis, fatty acid oxidation, and glucose transport. Because of all this, during adulthood these changes can contribute to susceptibility to obesity, type II diabetes, and metabolic syndrome [8]. In addition, there are differences at the haematological and biochemical level. Amdi et al. [7] explain how intrauterine growth restriction in piglets alters blood cell counts. Piglets with IUGR had a lower percentage of lymphocytes and erythrocytes than normal piglets. Further, the levels of mean corpuscular haemoglobin (MCH), mean cell volume (MCV), percentage neutrophils, and reticulocytes in IUGR pigs were higher than those in normal pigs. Based on these findings, it is possible that macrocytic anaemia results from compensatory mechanisms in IUGR pigs. Moreover, IGF-1 levels were similar in IUGR and normal pigs, and blood biochemical profiles of piglets at day 24 were not different from those of normal piglets; these factors indicate similar protein turnover, enzyme activity, and growth in the two phenotypes [7]. However, other blood parameters in piglets with uterine growth restriction have been poorly described or are unknown.

Piglets with IUGR characteristics may exhibit differences in their haematological and biochemical parameters at 72 h that may affect their growth and development compared to piglets born with normal weight. The objective of this study is to increase knowledge of the haematological and biochemical differences between normal piglets and IURG piglets for this purpose in addition to analytes previously measured; other new analytes such as thiol, CUPRAC, and insulin that can provide information about the metabolism and redox status of the animals were evaluated.

## 2. Materials and Methods

### 2.1. Facilities/Housing

The study was conducted on a commercial farm located in Pulpí (Almería, Southeastern Spain) with 2000 Large White × Landrace sows. The sows were inseminated with Duroc × Piétrain sperm and housed in individual pens (2.5 m long × 0.65 m wide) until gestation was confirmed. Then, they were reallocated to pens with a minimum space of 2.5 m^2^ per animal, in compliance with the European Union regulations concerning the protection of animals used for experimental and other scientific purposes [9]. Five days before farrowing, the sows were moved to farrowing rooms and placed in individual crates (2.36 m long × 1.5 m wide), where they stayed until the end of lactation—about 24 days. The piglets’ area was heated using a heating lamp. The ambient temperature in the farrowing unit was maintained at approximately 21 °C, and ventilation was thermostatically regulated. All the sows were fed a common cereal–soybean meal-based diet as per the Spanish Foundation for the Development of Animal Nutrition recommendations for sows during gestation and lactation [10]. The composition of the diet for a gestating sow and its nutritional levels are show in Table 1.

### 2.2. Animals and Experimental Design

Forty piglets from thirty-three multiparous sows (parity ranging from two to six) housed in the same farrowing room and with similar expected farrowing dates were selected for this trial. Immediately after farrowing, 20 piglets weighing more than 1500 g were selected and assigned to a control group (N) and 20 piglets with a body weight between 500 and 1000 g and with at least two IUGR features were assigned to a IUGR group (IUGR). The characteristics considered were those described by Engelsmann et al. [3]: steep dolphin-shaped forehead, bulging eyes, and hair without a single direction of growth. Piglets were individually identified by an ear tag and kept with their own sows to ensure colostrum suckling. Cross-fostering was promoted to standardize litters to 14 piglets at 48 h after birth. All experimental piglets were handled similarly, receiving an intramuscular dose of 532.6 mg of gleptoferron (200 mg of Fe^3+^) at 48 h of age (Gleptafer 200, SYVA, S.A.U, León, Spain).

### 2.3. Sampling

At 72 h after birth, ensuring colostrum feeding of all piglets, a blood sample (2 mL) was collected from each piglet by jugular venepuncture using a hypodermic needle (0.6 × 25 mm, 23 G) and a plastic syringe. Once collected, blood samples were transferred into heparinized tubes (Vacutainer, Becton, Dickinson, and Company, Franklin Lakes, NJ, USA) and an aliquot was centrifuged at 2000× *g* for 10 min to collect plasma, which was frozen at −80 °C until further analysis. The rest of the sample was used for haematological analysis.

### 2.4. Analysis

Blood samples were transported by refrigeration from farm to the Interdisciplinary Laboratory of Clinical Analysis of the University of Murcia (Interlab-UMU, Murcia, Spain) and immediately processed. Heparin blood samples were analysed for the following haematologic analytes with an automated haematological analyser (ADVIA 120, Siemens Healthcare Diagnostics SL, Barcelona, Spain): red blood cell count (RBC), haematocrit (HCT), haemoglobin (HB), mean corpuscular volume (MCV), mean corpuscular haemoglobin (MCH), mean corpuscular haemoglobin concentration (MCHC), cellular haemoglobin content (CH), corpuscular haemoglobin concentration mean (CHCM), cellular haemoglobin distribution width (CHDW), erythrocyte distribution width (RDW), and haemoglobin concentration distribution width (HDW). Reticulocyte indices were also determined, including reticulocyte count (Ret), reticulocyte haemoglobin content (CHr), average size of reticulocytes (MCVr), average cell haemoglobin concentration of reticulocytes (CHCMr), cellular haemoglobin distribution width of reticulocytes (CHDWr), distribution width of reticulocyte cell size (RDWr), distribution width of CHCMr (HDWr), microcytic reticulocytes (Micro-r), macrocytic reticulocytes (Macro-r), hypochromic reticulocytes (Hypo-r), hyperchromic reticulocytes (Hyper-r), reticulocytes with a low CH (Low CHr), and reticulocytes with a high CH (High CHr). Total white blood cell count (WBC), neutrophiles, lymphocytes, monocytes, eosinophiles, and basophiles were also counted. Platelet indices were analysed and included platelet count (PLT), mean platelet volume (MPV), plateletcrit (PCT), platelet volume distribution width (PDW), PLT component (MPC), mean platelet component distribution width (PCDW), mean platelet mass (MPM), platelet mass distribution width (PMDW), and large PLT.

Evaluation of the general metabolic profile in plasma samples (unbound iron binding capacity (UIBC)), albumin, amylase, total cholesterol, alkaline phosphatase (ALP), gamma-glutamyl transferase (GGT), glucose, aspartate aminotransferase (AST), alanine aminotransferase (ALT), iron, lipase, triglycerides (TG), lactate, uric acid, thiol, and lactate dehydrogenase (LDH)) was performed using an automatic analyser (Olympus AU400; Olympus, Tokyo, Japan). Cortisol was analysed with an automate chemiluminescent immunoassay (Immulite System, Siemens Health Diagnostics, Deerfield, IL, USA; detection limit 0.2 μg/dL). Insulin was determined by Enzyme-Linked Immuno Sorbent Assay (ELISA) Kit (Mercodia, Porcine Insulin ELISA, Winston Salem, NC, USA; reference 10-1200-01; detection limit ≤ 1.15 mU/L).

Antioxidant capacity was measured by the CUPRAC method (cupric ion reducing antioxidant capacity), as described by Contreras et al. [12]. Paraoxonase activity (PON1) was determined as described by Escribano et al. [13].

Concentrations of pig IgG, IgA, and IgM were assessed using a specific ELISA quantification kit purchased from Bethyl Laboratories, Inc. (Montgomery, TX, USA; references E100-104, E100-102, and E100-117; assay range 500–7.8 ng/mL, 1000–15.6 ng/mL, and 1000–15.6 ng/mL, respectively). The assay was carried out according to the manufacturer’s instructions.

The assay showed intra-assay coefficients of variation (CVs) below 15%.

### 2.5. Statistical Analysis

Statistical analyses were performed using the SPSS statistics package (IBM SPSS Statistics for Windows, Version 15.0. IBM Corp., New York, NY, USA). Data were assessed for normality using the Shapiro–Wilk method and those that showed non-normal distribution were log-transformed before being evaluated with a Student’s *t*-test. Probability values less than 0.05 (*p* < 0.05) were considered a significant difference.

## 3. Results

The number of piglets born to the sows used in the study was 15.2, of which 13.3 were live-born piglets and 1.9 were stillborn and mummified piglets.

### 3.1. Haematological Variables

#### 3.1.1. Red Blood Cell Parameters

Red blood cell parameters are shown in Table 2. RBC, HCT, HB, and RDW were lower in the IUGR group (*p* < 0.05). On the other hand, HDW was significantly higher in the same piglet group. The rest of the parameters showed no statistically significant differences.

#### 3.1.2. Reticulocyte Indices

The results of the reticulocytes’ indices showed statistically significant differences in some parameters between the N and IUGR groups (Table 3). Ret, CHr, MCVr, Macro-r, and High CHr were higher in the N group. However, RDWr, HDWr, and Low CHr were higher in the IUGR group. Statistically significant differences were not observed for the remaining parameters.

#### 3.1.3. White Blood Cell Parameters

White cell parameters in the N and IUGR groups are presented in Table 4. Lymphocytes (×10^3^/µL) were significantly lower in the IUGR group with respect to the N group. However, eosinophil’s percentage was significantly higher in the IUGR group compared to the N group. The other parameters analysed showed no significant differences between the two groups.

#### 3.1.4. Platelet Indices

Platelet indices are shown in Table 5. MPV, PCT, and large PLT were significantly higher in IUGR piglets than in normal piglets. The rest of the platelet indices were similar in the two groups (*p* > 0.05).

### 3.2. Biochemical Parameters

Some biochemical parameters (Table 6) showed statistically significant differences between normal and IUGR piglets. Thiol and CUPRAC were lower in the IUGR group. Nevertheless, insulin was higher in the IUGR group. All other parameters were not significantly affected.

### 3.3. Plasma Ig Levels

Plasma immunoglobulin (IgG, IgA, IgM) levels 72 h after birth do not differ significantly from the normal or from IUGR piglets (Table 7).

## 4. Discussion

With this study, we sought to understand and provide information about the changes in haematological and biochemical parameters that can occur in piglets suffering from IUGR within hours of birth versus those considered as normal. Research on blood parameters in pigs is not very profuse [14], with few blood parameters established in newborn piglets. The bibliography consulted refers mainly to piglets almost one month old [7,15].

The effects of birth weight on reference blood values were studied in 1- and 21-day-old piglets by Cincović et al. [16]. These authors observed that heavier piglets have a higher blood volume and higher haematocrit and haemoglobin levels, which have a positive effect at later stages, due to increasing daily intake at weaning, given that red blood cell parameters play a crucial role in piglet growth and weight gain [16]. In our study, IUGR piglets with lower weight than normal piglets showed lower values of RBC, HCT, and HB, in line with the observations by Cincovic et al. [16]. In addition, these differences have been observed after three weeks of age [17]. RBC formation and maturation is dependent on organ development [18], so the RBC value could be significantly lower in IUGR piglets because of their commitment to growth [4] or other factors, and therefore it needs more detailed investigations.

Furthermore, when assessing red blood cell variables, reticulocyte indices should also be considered. The parameters are closely related, as reticulocytes correspond to the immature forms of red blood cells. Some reticulocyte indices, such as CHr, are a sign of iron shortage [19]. However, no significant differences were observed between the two groups when the iron value, which is strongly associated with the reticulocyte indices and haemoglobin, was analysed. The haemoglobin value, despite being significantly lower in the IUGR group, was above 80 g/L, indicating no anaemia [18,20]. This may be explained by the dose of iron that piglets received as part of habitual farm practices to make up for the iron deficiency that this species exhibits at birth. Nevertheless, although haemoglobin levels are normal at birth, after 10–14 days old these levels drop rapidly [21]. Because of a reduction in haemoglobin, rapid growth, and iron shortages, among other causes, piglets are predisposed to hypochromic, microcytic anaemia (iron deficiency anaemia) [22]. Thus, the low RBC value coupled with lower levels of MCVr and CHr found in our report compared to normal pigs could indicate greater risk for the future development of hypochromic, microcytic anaemia in IUGR pigs.

Regarding reticulocyte maturation, maturing reticulocytes are found primarily in the bone marrow until they enter the bloodstream and complete the maturation phase [23]. Reticulocytes indicate recent bone marrow activity because of their brief lifespan in the circulatory system [19]. Quite different reticulocyte indices were observed between the two study groups. These differences, along with the previously noted distinctions in red blood cells, could be indicative of differences in bone marrow function between normal piglets and IUGR piglets. Amdi et al. [7] also reported differences in the reticulocyte indices between the two groups of piglets at 24 days old and unusual bone marrow haematopoiesis in IUGR pigs.

Through the results of white blood cell parameters, we can obtain information about the immune system, one of the most studied traits of IUGR piglets. IUGR piglets show a reduced response of blood leucocytes to infection, which increases the susceptibility of these piglets to illness [24]. Baek et al. [25] explained the low leukocytes number by a decrease in lymphocyte counts found in IURG piglets, in agreement with our findings at 72 h. Despite the lymphocyte decrease, none of the other leukocyte proportions was different, except for eosinophils. The lack of differences in the other parameters could be due to undeveloped immune systems of all piglets at birth [26]. The eosinophil percentage was higher in IUGR piglets, which could be caused by hypoxia [27]. It is known that IUGR piglets suffer inadequate perfusion [28]. Further, these piglets frequently belong to large litters, where farrowing duration is prolonged, increasing the hypoxia risk [29]. In addition, an increase in eosinophils could be due to a parasitic infection [30].

According to the platelet indices, all parameters were generally found to match the reference values estimated by Ventrella et al. [31] for five-day-old piglets. However, the values obtained for PLT were lower in our groups of piglets. The intensive growth and maturation of the haematopoietic system may explain the PLT variability in young pigs [32]. MPV was found to be higher in IUGR piglets, in line with the values established by Baek et al. [25] in 8- to 10-day-old piglets. As an indicator of platelet activation and production rate, MPV has some clinical relevance, and an increased MPV indicates an increase in platelet size [33]. However, when interpreting this parameter, one must consider which MPV measurement methods have been used, as different techniques can produce results that vary by up to 40% [34]. Nevertheless, it is important to mention that platelet indices act as inflammatory markers in various intestinal conditions [35] and in IUGR piglets they have been described as intestinal inflammatory damage [36]. Therefore, the increase observed in some platelet indices in our IUGR piglets group respective to the normal group could be due to this fact.

Concerning biochemical parameters, the IUGR group showed a noticeable rise in insulin. Intrauterine growth restriction is associated with insulin resistance, so elevated levels of this hormone would be related to this fact [37]. In this situation, blood insulin levels are not enough to manage metabolic processes [37]; as a result, compensatory hyperinsulinemia ensues [38]. Insulin resistance is stimulated by inflammation in the liver [39], and IUGR is linked to foetal liver inflammation, which has been described in IUGR piglets [40]. Disturbances in insulin secretion and insulin resistance are factors related to type II diabetes [41]. Several authors have described in humans an association between IUGR and type II diabetes, as well as metabolic syndrome [42,43]. The metabolic syndrome makes up for various risk factors, such as type II diabetes and cardiovascular disease [42], and it is characterized by insulin resistance, hyperinsulinemia, elevated triglycerides, fatty liver, hypertension, renal failure, and inflammation, among other clinical indicators [41]. Also, Shen at al. [44] described higher levels of glucose and triglycerides in adult IUGRs than in the normal group. However, in the present study no significant differences were observed between the two groups of piglets for these parameters, probably because it was an incipient process and these parameters’ values were not high enough to show a discrepancy.

With respect to the oxidative status, IUGR piglets have lower levels of thiol and CUPRAC than normal piglets. In an imbalanced situation, antioxidants and protective mechanisms in the body, are unable to produce and dispose of reactive oxygen species (ROS), resulting in oxidative stress [45]. Several authors have indicated the close relationship between IUGR and oxidative stress [36,40,46], and these observations are extended to the weaning stage [45]. The placental hypoxia and inadequate perfusion suffered by IUGR piglets are a possible reason for oxidative stress [28]. This oxidative imbalance situation is related to various illnesses and clinical disorders, so these piglets would be more susceptible to future diseases [47]. Therefore, IUGR piglets are predisposed to metabolic disorders, such as the metabolic syndrome mentioned above, due to the presence of many of these conditions because of oxidative stress [45].

On the other hand, it should be noted that several stressors are suffered by piglets belonging to large litters due to competition for colostrum and conflicts over the udder; as a result, cortisol levels rise [48]. Also, IUGR piglets have less energy than their non-IUGR peers in competing for the udder [3], and maternal space constraint increases foetal glucocorticoids [49]. Thus, in the IUGR group, a numerical increase in this stress marker was observed with respect to the normal group. Furthermore, lymphocytes, which are present in lower number in IUGR piglets, could be a sign that reaffirms the piglets’ stressed status [50].

Finally, immunoglobulins, an essential component of the piglet’s immune system, combined with the antibodies produced by B cells, indicate the level of humoral immunity [51]. It was observed that IgG was the most common isotype found in piglet serum; this is also consistent with the fact that it is the predominant immunoglobulin in sow colostrum [52]. Piglets are born agammaglobulinaemic due to the epitheliochorial nature of the porcine placenta, which makes ingesting colostrum the only way for the piglet to acquire immunoglobulins [26]. Therefore, the IgG concentration in the piglet’s serum will depend on the IgG levels in the sow’s colostrum. Lower-weight piglets have lower colostrum intake [53]; consequently, lower levels of immunoglobulins would be expected in the IUGR group. However, in our case, no differences were observed. In addition, according to the literature, mean blood IgG levels in pigs during the first days of life are greater than those discovered in our investigation. Lazarevic et al. [54] established a mean serum IgG concentration in piglets at 48 h of 41.95 mg/mL and 49.52 mg/mL in two studies conducted. One of the possible explanations for this low serum IgG concentration could be insufficient colostrum or a low IgG concentration in the maternal colostrum. Colostrum production in sows is independent of litter size; thus, piglets from larger litters have less colostrum available [55], and the sows in our study were hyperprolific. In summary, the current study found that immune system alterations in IUGR piglets are mostly due to the white blood cell parameters rather than changes in the immunoglobulins.

## 5. Conclusions

At 72 h postpartum, IUGR and, therefore, underweight piglets are haematologically and biochemically different from normal piglets. Reduced RBC, HCT, and HB, together with altered reticulocyte indices, suggest the possible development of microcytic hypochromic anaemia. In addition, IURG piglets show low lymphocyte counts that could be influenced by the high cortisol levels. From a biochemical point of view, hyperinsulinemia in IUGR piglets, together with the oxidative imbalance that they exhibit, makes them more vulnerable to disease and the development of future disorders, such as metabolic syndrome. However, at the immunoglobulin level, no differences were observed between the two groups.

## Figures and Tables

**Table 1 animals-13-03540-t001:** Diet composition (%) and nutritional levels (%, as fed basis) for gestating sows.

Item	Diet
Barley	51.03
Wheat	18.43
Wheat bran	9.52
Sunflower meal 28%	10.00
Soybean meal 47%	1.00
Beet pulp	7.00
Lard	0.40
Calcium carbonate	1.07
Monocalcium phosphate	0.23
Salt	0.45
L-Lysine HCl, 78%	0.24
L-Threonine	0.08
Premix ^1^	0.55
Calculated nutrient composition (%, as fed basis) ^2^	
Dry matter	89.23
Crude ash	4.71
Crude protein	12.60
Crude fat	2.04
Neutral detergent fiber	21.27
Acid detergent fiber	8.88
Starch	40.46
SID lysine ^3^	0.56
Net Energy, kcal/kg	2275

^1^ Premix provided vitamins and minerals per kg diet as follow: vitamin A (3a672a), 10,000 IU; vitamin D3 (3a671), 1800 IU; vitamin E (3a700), 60 IU; vitamin K3 (3a711), 1.5 mg; vitamin B1 (3a821), 1.5 mg; vitamin B2 (3a825ii), 5 mg; vitamin B6 (3a831), 2.5 mg; vitamin B12, 0.05 mg; niacin (3a315), 25 mg; pantothenic acid (3a841), 15 mg; folic acid (3a316), 2.5 mg; biotin (3a880), 0.3 mg; Mn (3b503), 32 mg and Mn (3b506) 8 mg; Zn (3b605), 60 mg and Zn (3b607) 40 mg; Cu (3b413), 10 mg; I (3b201), 0.80 mg; Se (3b801), 0.1 mg and Se (3b814), 0.2 mg; Fe (3b103), 64 mg and Fe (3b108), 16 mg; 6-phytase (EC 3.1.3.26), 1000 FTU; sepiolite (E562), 300 mg; butylated hydroxyanisole (E320), 0.06 mg; butylated hydroxytoluene (E321), 0.24 mg; *Bacillus velezensis* (4b1820) 3.0 × 10^8^ UFC; citric acid (1a330), 0.10 mg; formic acid (1k236), 390 mg; lactic acid (1a270), 100 mg; ammonium formate (1a295), 150 mg; ammonium propionate (1k284), 185 mg. ^2^ According to the Spanish Foundation for the Development of Animal Nutrition [11]. ^3^ SID lysine = Standardized ileal digestible lysine.

**Table 2 animals-13-03540-t002:** Differences between normal piglets and IUGR piglets at birth on red blood cell variables. Data are expressed as mean ± SD.

Variable	N	IUGR	*p*-Value
RBC (×10^6^/µL)	4.61 ± 0.47	3.95 ± 0.53	0.001
HCT (%)	30.02 ± 2.6	25.79 ± 2.9	0.001
HB (g/dL)	9.60 ± 0.76	8.60 ± 1.22	0.008
MCV (fL)	65.24 ± 2.90	65.24 ± 3.03	0.997
MCH (pg)	20.88 ± 1.15	21.84 ± 2.39	0.145
MCHC (g/dL)	32.01 ± 0.99	33.49 ± 3.55	0.105
CH (pg)	19.35 ± 0.88	19.39 ± 1.05	0.901
CHCM (g/dL)	29.91 ± 0.65	29.86 ± 0.46	0.822
CHDW (pg)	3.47 ± 0.25	3.33 ± 0.26	0.154
RDW (%)	22.87 ± 2.90	20.30 ± 2.32	0.011
HDW (g/dL)	3.22 ± 0.25	3.46 ± 0.34	0.030

N: normal group, piglets with weight at birth > 1500 g; IUGR: intrauterine growth restriction group, piglets weighing between 500 and 1000 g with at least two IUGR features; RBC: red blood cell count; HCT: haematocrit; HB: haemoglobin; MCV: mean corpuscular volume; MCH: mean corpuscular haemoglobin; MCHC: mean corpuscular haemoglobin concentration; CH: cellular haemoglobin content; CHCM: corpuscular haemoglobin concentration mean; CHDW: cellular haemoglobin distribution width; RDW: erythrocyte distribution width; HDW: haemoglobin concentration distribution width.

**Table 3 animals-13-03540-t003:** Differences between normal piglets and IUGR piglets at birth in terms of reticulocyte indices. Data are expressed as mean ± SD.

Variable	N	IUGR	*p*-Value
Ret (10^6^/µL)	0.44 ± 0.12	0.29 ± 0.11	0.001
CHr (pg)	22.45 ±1.41	21.01 ± 2.38	0.042
MCVr (fL)	83.48 ± 4.16	78.54 ± 7.62	0.026
CHCMr (g/dL)	27.11 ± 0.58	27.08 ± 0.64	0.902
CHDWr (pg)	3.76 ± 0.32	3.69 ± 0.IL46	0.628
RDWr (%)	16.74 ± 1.67	18.19 ± 1.96	0.032
HDWr (g/dL)	3.44 ± 0.32	4.05 ± 0.63	0.001
Micro-r (%)	0.72 ± 0.51	0.79 ± 0.48	0.669
Macro-r (%)	58.76 ± 14.13	45.11 ± 21.71	0.040
Hypo-r (%)	68.01 ± 7.40	67.01 ± 6.61	0.695
Hyper-r (%)	0.62 ± 0.41	0.90 ± 0.47	0.086
Low CHr (%)	1.82 ± 0.81	4.10 ± 4.47	0.032
High CHr (%)	64.27 ± 14.75	49.71 ± 24.36	0.045

N: normal group, piglets with weight at birth > 1500 g; IUGR: intrauterine growth restriction group, piglets weighing between 500 and 1000 g with at least two IUGR features; Ret: reticulocyte count; CHr: reticulocyte haemoglobin content; MCVr: average size of reticulocytes; CHCMr: average cell haemoglobin concentration of reticulocytes; CHDWr: cellular haemoglobin distribution width of reticulocytes; RDWr: distribution width of reticulocyte cell size; HDWr: distribution width of CHCMr; Micro-r: microcytic reticulocytes; Macro-r: macrocytic reticulocytes; Hypo-r: hypochromic reticulocytes; Hyper-r: hyperchromic reticulocytes; Low CHr: reticulocytes with a low CH; High CHr: reticulocytes with a high CH.

**Table 4 animals-13-03540-t004:** Differences between normal piglets and IUGR piglets at birth on white cell variables. Data are expressed as mean ± SD.

Variable	N	IUGR	*p*-Value
WBC (×10^3^/µL)	10.06 ± 3.57	8.34 ± 2.69	0.145
Neutrophil (%)	52.32 ± 13.34	58.19 ± 8.06	0.157
Neutrophil (×10^3^/µL)	5.56 ± 3.41	4.82 ± 1.65	0.466
Lymphocite (%)	41.35 ± 12.41	35.78 ± 8.72	0.165
Lymphocite (×10^3^/µL)	3.90 ± 0.98	3.02 ± 1.41	0.046
Monocyte (%)	2.43 ± 1.47	2.89 ± 1.55	0.402
Monocyte (×10^3^/µL)	0.23 ± 0.11	0.24 ± 0.14	0.888
Eosinophil (%)	0.64 ± 0.45	0.96 ± 0.35	0.032
Eosinophil (×10^3^/µL)	0.06 ± 0.04	0.08 ± 0.04	0.169
Basophile (%)	0.52 ± 0.21	0.49 ± 0.38	0.733
Basophile (×10^3^/µL)	0.05 ± 0.02	0.04 ± 0.02	0.194

N: normal group, piglets with weight at birth > 1500 g; IUGR: intrauterine growth restriction group, piglets weighing between 500 and 1000 g with at least two IUGR features; WBC: total white blood cell count.

**Table 5 animals-13-03540-t005:** Differences between normal piglets and IUGR piglets at birth on platelet indices. Data are expressed as mean ± SD.

Variable	N	IUGR	*p*-Value
PLT (×10^3^/µL)	188.11 ± 102	230.57 ± 82.29	0.215
MPV (fL)	11.69 ± 2.48	14.66 ± 4.62	0.023
PCT (%)	0.23 ± 0.11	0.34 ± 0.15	0.026
PDW (%)	70.37 ± 15.40	77.45 ± 5.95	0.137
MPC (g/dL)	23.22 ± 1.00	23.79 ± 0.85	0.103
PCDW (g/dL)	6.43 ± 0.76	6.14 ± 0.58	0.254
MPM (pg)	1.99 ± 0.24	2.07 ± 0.14	0.272
PMDW (pg)	0.95 ± 0.06	0.97± 0.04	0.303
Large PLT (×10^3^/µL)	23.06 ± 10.95	44.00 ± 31.57	0.022

N: normal group, piglets with weight at birth > 1500 g; IUGR: intrauterine growth restriction group, piglets weighing between 500 and 1000 g with at least two IUGR features; PLT: platelet count; MPV: mean platelet volume; PCT: plateletcrit; PDW: platelet volume distribution width; MPC: PLT component; PCDW: mean platelet component distribution width; MPM: mean platelet mass; PMDW: platelet mass distribution width.

**Table 6 animals-13-03540-t006:** Differences between normal piglets and IUGR piglets at birth in terms of plasma biochemistry. Data are expressed as mean ± SD.

Variable	N	IUGR	*p*-Value
TIBC (µg/dL)	434 ± 236	650 ± 472	0.420
Iron (µg/dL)	423.4 ± 268	658.2 ± 515	0.104
UIBC (µg/dL)	65.72 ± 63.88	63.88 ± 61.62	0.214
Albumin (g/dL)	1.24 ± 0.15	1.32 ± 0.25	0.233
Amylase (UI/L)	1239 ± 381	1257 ± 466	0.900
Total Cholesterol (mg/dL)	130.2 ± 36.5	115.59 ± 29.86	0.180
ALP (UI/L)	1914.1 ± 625	2278.9 ± 1019	0.189
GGT (UI/L)	71.5 ± 20.58	66.9 ± 7.30	0.364
Glucose (mg/dL)	94.94 ± 18.33	102.54 ± 14.49	0.163
AST (UI/L)	60.1 ± 15.92	77.1 ± 25.05	0.495
ALT (UI/L)	30.89 ± 28.48	26.93 ± 30.81	0.903
Lipase (UI/L)	74.11 ± 31.01	83.71 ± 33.69	0.361
TG (mg/L)	164.9 ± 76.34	236.6 ± 189	0.134
Insulin (UI/L)	15.58 ± 8.82	34.50 ± 23.78	0.002
Lactate (mmol/L)	8.09 ± 2.39	7.85 ± 1.07	0.685
Uric acid (mg/dL)	0.223 ± 0.12	0.250 ± 0.10	0.465
Thiol (mmol/L)	0.063 ± 0.29	0.032 ± 0.032	0.004
CUPRAC (mmol/L)	0.272 ± 0.055	0.224 ± 0.030	0.002
LDH (UI/L)	1747.8 ± 380	1617.7 ± 257	0.243
PON1 (UI/mL)	2.57 ± 1.17	1.95 ± 1.25	0.114
Cortisol (µg/dL)	4.89 ± 2.87	7.05 ± 5.02	0.112

N: normal group, piglets with weight at birth > 1500 g; IUGR: intrauterine growth restriction group, piglets weighing between 500 and 1000 g with at least two IUGR features; UIBC: unbound iron binding capacity platelet count; ALP: alkaline phosphatase; GGT: gamma-glutamyl transferase; AST: aspartate aminotransferase; ALT: alanine aminotransferase; TG: triglycerides; LDH: lactate dehydrogenase; PON1: paraoxonase activity.

**Table 7 animals-13-03540-t007:** Differences between normal piglets and IUGR piglets at birth in terms of immunological variables. Data are expressed as mean ± SD.

Variable	N	IUGR	*p*-Value
IgG (mg/mL)	25.87 ± 12.50	21.44 ± 9.00	0.211
IgA (mg/mL)	5.37 ± 3.17	4.13 ± 2.32	0.170
IgM (mg/mL)	1.94 ± 0.81	2.72 ± 1.61	0.069
Ig total	33.18 ± 14.51	28.28 ± 11.19	0.244

N: normal group, piglets with weight at birth > 1500 g; IUGR: intrauterine growth restriction group, piglets weighing between 500 and 1000 g with at least two IUGR features.

## Data Availability

The data presented in this study are available on request from the corresponding author (S.M.-M.).

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
