# Peer review of "A Comparison of Haematological and Biochemical Profiles between Intrauterine Growth Restriction and Normal Piglets at 72 Hours Postpartum"

_animals, 2023, doi:10.3390/ani13223540_

Round 1

Reviewer 1 Report

Comments and Suggestions for Authors

The manuscript entitled “Comparison of Haematological and Biochemical Profiles between Intrauterine Growth Restriction and Normal Piglets at 72 Hours Postpartum” evaluated haematological and biochemical parameters as well as immunoglobulin levels of intrauterine growth restricted piglets and compared the values with normal piglets. The topic of the current manuscript is very interesting and addresses a practical problem of pig industry.  

As prolificacy is an important economic trait in swine industry,  most breeding programs are trageted towards increasing the litter size of breeding sows. By genetic selection and selective breedibg, litter size in sows has increased dramatically in recent decades. One of the negative consequences of this is piglets born with intrauterine growth restriction (IUGR). The authors mentioned that around 30% of newborn pigs in Denmark showed signs of IUGR which is alarming. Between normal and  IUGR piglets, there are several physiological differences including haematological and biochemical changes. Some reports are already available on this. But I agree with the authors that detailed picture of blood parameters in piglets with uterine growth restriction are unknown. In that context, the current study addresses an existing knowledge gap.

 The work is important and well presented. The writing is also very good. I do not have any concern with the current manuscript.

 The composition of the diet and its nutritional levels may be provided.

Comments on the Quality of English Language

It is good expect some minor typos.

Author Response

Dear Reviewer 1.

Thank tou for your suggestion for improve our work. I respond to your comment

The manuscript entitled “Comparison of Haematological and Biochemical Profiles between Intrauterine Growth Restriction and Normal Piglets at 72 Hours Postpartum” evaluated haematological and biochemical parameters as well as immunoglobulin levels of intrauterine growth restricted piglets and compared the values with normal piglets. The topic of the current manuscript is very interesting and addresses a practical problem of pig industry.  

As prolificacy is an important economic trait in swine industry,  most breeding programs are trageted towards increasing the litter size of breeding sows. By genetic selection and selective breedibg, litter size in sows has increased dramatically in recent decades. One of the negative consequences of this is piglets born with intrauterine growth restriction (IUGR). The authors mentioned that around 30% of newborn pigs in Denmark showed signs of IUGR which is alarming. Between normal and  IUGR piglets, there are several physiological differences including haematological and biochemical changes. Some reports are already available on this. But I agree with the authors that detailed picture of blood parameters in piglets with uterine growth restriction are unknown. In that context, the current study addresses an existing knowledge gap.

The work is important and well presented. The writing is also very good. I do not have any concern with the current manuscript.

The composition of the diet and its nutritional levels may be provided.

Response: Thank you for your comments. The composition of the diet and its nutritional levels has been provided in Table 1.

Reviewer 2 Report

Comments and Suggestions for Authors

In the present article Authors characterise the haematological and biochemical blood parameters in the piglets with intrauterine growth restriction  at 72 hours after birth. Topic of the study is interesting and timely and results interesting because about 30% of piglets born with low birth weight. However the following issues that arose during the evaluation of the manuscript should be addressed.            Introduction Line 82-83. Authors claim that “Proteomic analysis in IUGR piglets’ intestines showed a decrease in the concentration of proteins implicated in iron metabolism, such as transferrin and retinol binding protein 2 (RBP2)” – RBP2 is not a protein related to iron metabolism.   Materials and Methods What was the size of the litters from which the IUGR piglets came?   Authors collected blood samples at 72 hour of birth and then analysed blood parameters in piglets with uterine growth restriction. However piglets from both investigated groups were treated with intramuscular dose of 532.6 mg of gleptoferron at 48 h of age and it can influence  on blood parameters of piglets. Moreover the piglets from both groups, despite of differences in body weight, received the some dosage of the gleptoferron. In my opinion Authors should analyse first all this blood parameters in the groups of intact piglets, because this would allow Authors to determine how low birth weight actually affects blood morphology and biochemical parameters. Of course such results could be compared with those obtained by Authors in the present study.   My second question is, how much iron is contained in 532.6 mg of gleptoferron? This information should be included in manuscript.     Results                                                                In Table 5 Authors showed iron concentration in the serum of the both analysed group of piglets treated with gleptoferron and I have noticed that iron concentration in the serum of the  IUGR piglets was high in comparison with control but differences was not significant because of very high SD value. Therefore it would be better to analyse TIBC because it can show how much iron is bound to transferrin and can be transported to the bone marrow. Such information would be very interesting and valuable because in the group of the IUGR piglets reticulocyte level was lower than in control group.   Discussion Line 275-276 According to the Authors “…. heavier piglets have a higher blood volume and, therefore, higher haematocrit and haemoglobin level…..” I can not agree with this statement, but probably the Authors wanted to say that“…. heavier piglets have a higher blood volume and higher haematocrit and haemoglobin level…..”.   Line 282-283. I can not agree with the Authors statement that in the IUGR group “….the RBC value could be significantly lower in IUGR piglets because of their commitment to growth”.  Lower RBC value in this group of piglets can be caused by many factors and it need more detailed investigations.

Author Response

Dear Reviewer 2.

Thank you for your suggestions to improve our work. Here are my responses to your comments.

In the present article Authors characterise the haematological and biochemical blood parameters in the piglets with intrauterine growth restriction  at 72 hours after birth. Topic of the study is interesting and timely and results interesting because about 30% of piglets born with low birth weight. However the following issues that arose during the evaluation of the manuscript should be addressed.    

Introduction

Line 82-83. Authors claim that “Proteomic analysis in IUGR piglets’ intestines showed a decrease in the concentration of proteins implicated in iron metabolism, such as transferrin and retinol binding protein 2 (RBP2)” – RBP2 is not a protein related to iron metabolism.  

Response: Thank you for your comment, we have corrected the sentence and this information has been included in the text. Line 88-89. Proteomic analysis in IUGR piglets’ intestines showed a decrease in the concentration of proteins implicated in iron metabolism, such as transferrin, and other proteins like retinol binding protein 2 (RBP2).     

Materials and Methods

What was the size of the litters from which the IUGR piglets came? 

Response: The number of piglets born to the sows used in the study was 15.2, of which 13.3 were live-born piglets and 1.9 were stillborn and mummified piglets. Line 215.

Authors collected blood samples at 72 hour of birth and then analysed blood parameters in piglets with uterine growth restriction. However piglets from both investigated groups were treated with intramuscular dose of 532.6 mg of gleptoferron at 48 h of age and it can influence  on blood parameters of piglets. Moreover the piglets from both groups, despite of differences in body weight, received the some dosage of the gleptoferron. In my opinion Authors should analyse first all this blood parameters in the groups of intact piglets, because this would allow Authors to determine how low birth weight actually affects blood morphology and biochemical parameters. Of course such results could be compared with those obtained by Authors in the present study.  

My second question is, how much iron is contained in 532.6 mg of gleptoferron? This information should be included in manuscript.

Response: All experimental piglets were handled similarly, receiving an intramuscular dose of 532.6 mg of gleptoferron (200 mg of Fe3+) at 48 h of age (Gleptafer 200, SYVA, S.A.U, León, Spain). This information has been included in the text. Line 151.

Results

In Table 5 Authors showed iron concentration in the serum of the both analysed group of piglets treated with gleptoferron and I have noticed that iron concentration in the serum of the  IUGR piglets was high in comparison with control but differences was not significant because of very high SD value. Therefore it would be better to analyse TIBC because it can show how much iron is bound to transferrin and can be transported to the bone marrow. Such information would be very interesting and valuable because in the group of the IUGR piglets reticulocyte level was lower than in control group.   

Response: Thank you for your comment, TIBC has been included in Table 6.

Discussion

Line 275-276 According to the Authors “…. heavier piglets have a higher blood volume and, therefore, higher haematocrit and haemoglobin level…..” I can not agree with this statement, but probably the Authors wanted to say that“…. heavier piglets have a higher blood volume and higher haematocrit and haemoglobin level…..”.

Response: Thank you for your suggestion, the text has been improved for better comprehension. Line 303.

Line 282-283. I can not agree with the Authors statement that in the IUGR group “….the RBC value could be significantly lower in IUGR piglets because of their commitment to growth”.  Lower RBC value in this group of piglets can be caused by many factors and it need more detailed investigations.

Response: Thank you for your suggestion, the text has been improved. Line 310.

Reviewer 3 Report

Comments and Suggestions for Authors

Manuscript 2657594 “Comparison of Haematological and Biochemical Profiles between Intrauterine Growth Restriction and Normal Piglets at 72 Hours Postpartum”

The knowledge on IUGR piglets is important in order to establish some possible management practices that can lead to a higher survival and growth of these piglets. The present study however, presents a major fault that makes impossible (in my opinion) a correct IUGR piglets characterization. Blood sampling was made at 72h and all colostrum/transition milk was not quantified (or at least estimated for the first 24h). The piglet intake during that period can, in my opinion, influence several of the analyzed parameters, therefore the results can be influenced by the piglet type but also by their feeding during the first 72h.

I write below some suggestions for the authors but because of that possible mixed influence on the results, I have to reject this manuscript.

Line 22 – The genetic selection is not “based on hyperprolificacy”, it is based on several traits. I advise to write “…genetic selection for hyperprolificacy”. Also replace “under-weight piglets” by “low birthweight piglets”

Line 63-68: suggestion: replace “essential that these piglets begin nursing as soon as possible in order to maintain their body temperature and gain energy. However, a lack of energy makes it difficult for them to compete for udder space and they will not have sufficient access to colostrum, resulting in a high rate of mortality [3]. Thus, there is no doubt that the size of a piglet at birth affects its chance of survival [1].”

by “essential that these piglets ingest colostrum as soon as possible in order to maintain their body temperature and increase their energy sources. However, the lack of energy at birth makes it difficult for them to compete for udder space and most of them will not have sufficient access to colostrum, resulting in a high mortality rate [3]. Thus, there is no doubt that the weight of a piglet at birth affects its chance to survive [1].

Line 70-71: Accordingly, lactate levels were higher in IUGR piglets. Your text in on the present, and then this sentence is in the past…..

Lines 80-81: “Due to the delay in digestive mucosa maturation, food changes and bacteria can be more difficult to adapt to [7].” I don’t understand this sentence….adapt to what?

Line 105: suggestion: replace “installations” by “facilities”

Line 108: replace Pietrain by Piétrain

Lines 116-118: If IUGR can be influenced by sow feeding during gestation, some details should be given here (feed energy, CP, amounts, etc)

Animals and Experimental Design (several questions):

Why “more than 1500g”? These piglets can be considered “normal”? What was the mean birth weight in those 33 litters (of all live-born piglets)?

All litters remained intact (with no piglet changes, added or removed, except if dead) in the first 24h?

What was the within-litter variability? The litter characteristics (number of piglets, weight variability, etc) can influence a lot the piglet’s performance. A 1kg weight piglet in a small litter, with a low mean weight and/or low weight variability can have a substantial higher probability to ingest a “sufficient” amount of colostrum, than the same piglet in a litter with the opposite characteristics….

No experimental piglet died before 72h? I think that it is rather strange that all the 20 piglets under 1kg at birth survived until 72h….

Why the piglets were not weighed at 24h? it could provide the estimation of the colostrum intake (using published prediction equations). Wouldn’t that be a fundamental information for all the analyzed traits (e.g. immunoglobulins levels)?

Line 143: what is the veterinary farm? It was not a commercial farm?

Line 179: you should provide the intra and inter-assay CV.

Results section:

You should give results on several traits such as: means of total and alive born piglets, the farrowing duration, the within-litter weight CV, etc, so that we can better analyze the results. Give results also regarding the experimental piglet’s performance, mortality rate, growth rate, weaning weight.

I don’t know the potential effects of colostrum on all the various studied parameters however I consider that for some of them, the amount of ingested colostrum could largely influence their values at 72h. For example, the Ig’s levels at 72h are highly influenced by colostrum intake and quality (composition) as colostrum is the only Ig’s source. In a “normal” situation IUGR piglets ingest much less colostrum (and probably with lower Ig levels) than piglets heavier than 1500g, however in your study no differences were observed. Additionally, besides Ig’s, colostrum is rich in maternal cells (mainly lymphocytes and epithelial cells) that can be absorbed by their offspring (Le Jan et al., 1995), so the colostrum intake could also influence lymphocytes levels.

Le Jan, C., Le Dividich, J., Chevaleyre, Hulin, J.C., 1995. Devenir des cellules colostrales chezle porc nouveau-né. Journées de la Recherche Porcine 27, 91–96.

Author Response

Dear Reviewer 3. 

Thank you for your suggestions to improve our work. Here are my responses to your comments

The knowledge on IUGR piglets is important in order to establish some possible management practices that can lead to a higher survival and growth of these piglets. The present study however, presents a major fault that makes impossible (in my opinion) a correct IUGR piglets characterization. Blood sampling was made at 72h and all colostrum/transition milk was not quantified (or at least estimated for the first 24h). The piglet intake during that period can, in my opinion, influence several of the analyzed parameters, therefore the results can be influenced by the piglet type but also by their feeding during the first 72h.

Response: Indeed, colostrum ingested at 24h was not quantified. The study of blood parameters was carried out at 72h to ensure that the piglets were correctly colostralised. As can be seen, there were no significant differences in immunoglobulin levels between the two groups, so that both groups were correctly colostrum feeding

I write below some suggestions for the authors but because of that possible mixed influence on the results, I have to reject this manuscript

Line 22 – The genetic selection is not “based on hyperprolificacy”, it is based on several traits. I advise to write “…genetic selection for hyperprolificacy”. Also replace “under-weight piglets” by “low birthweight piglets”.

Response: Thank you for your suggestion. You are right, we have changed the phrase for your better understanding. Line 23.

Line 63-68: suggestion: replace “essential that these piglets begin nursing as soon as possible in order to maintain their body temperature and gain energy. However, a lack of energy makes it difficult for them to compete for udder space and they will not have sufficient access to colostrum, resulting in a high rate of mortality [3]. Thus, there is no doubt that the size of a piglet at birth affects its chance of survival [1].”by “essential that these piglets ingest colostrum as soon as possible in order to maintain their body temperature and increase their energy sources. However, the lack of energy at birth makes it difficult for them to compete for udder space and most of them will not have sufficient access to colostrum, resulting in a high mortality rate [3]. Thus, there is no doubt that the weight of a piglet at birth affects its chance to survive [1]

Response: The text has been changed according to your suggestion. Line 65-69.

Line 70-71: Accordingly, lactate levels were higher in IUGR piglets. Your text in on the present, and then this sentence is in the past…..

Response: Thanks, this verb form has been homogenized. Line 77

Lines 80-81: “Due to the delay in digestive mucosa maturation, food changes and bacteria can be more difficult to adapt to [7].” I don’t understand this sentence….adapt to what?

Response: Thank you for your observation, this sentence has a mistake, the text has been corrected. Line 86.

Line 105: suggestion: replace “installations” by “facilities”

Response: Thank you, Installations has been changed by facilities. Line 112.

Line 108: replace Pietrain by Piétrain

Response: The change has been made. Line 115.

Lines 116-118: If IUGR can be influenced by sow feeding during gestation, some details should be given here (feed energy, CP, amounts, etc)

Response: Information about the composition of the diet and its nutritional value of sow has been added in Table 1.

Animals and Experimental Design (several questions):

Why “more than 1500g”? These piglets can be considered “normal”? What was the mean birth weight in those 33 litters (of all live-born piglets)?

Response: In this farm the average weight of piglets was 1347 ± 335g. To ensure their total normality and complete development were taken from the 65th percentile and without IUGR signs.

All litters remained intact (with no piglet changes, added or removed, except if dead) in the first 24h?

Response: Yes, the objective was to have the piglets take colostrum from their mother.

What was the within-litter variability? The litter characteristics (number of piglets, weight variability, etc) can influence a lot the piglet’s performance. A 1kg weight piglet in a small litter, with a low mean weight and/or low weight variability can have a substantial higher probability to ingest a “sufficient” amount of colostrum, than the same piglet in a litter with the opposite characteristics….

Response: The number of piglets born to the sows used in the study was 15.2, of which 13.3 were live-born piglets and 1.9 were stillborn and mummified piglets. Within-litter variability was 26.7%.

No experimental piglet died before 72h? I think that it is rather strange that all the 20 piglets under 1kg at birth survived until 72h….

Response: There were indeed deaths of underweight piglets in the first 72 h post parturition, on this farm 18% mortality was observed. This study is part of a larger project that collects much more information. However, in this experiment we only considered piglets that were alive at 72 h to obtain a blood sample.

Why the piglets were not weighed at 24h? it could provide the estimation of the colostrum intake (using published prediction equations). Wouldn’t that be a fundamental information for all the analyzed traits (e.g. immunoglobulins levels)?

Response: This would have been interesting information, but we did not consider it in the study. We confirmed that they had ingested colostrum by testing for immunoglobulins.

Line 143: what is the veterinary farm? It was not a commercial farm?

Yes, it was a commercial farm. There is a mistake in the text which has been corrected. Line 164.

Line 179: you should provide the intra and inter-assay CV.

Response: Thank your comment. The assay showed intra-assay coefficients of variation (CVs) below 15%. We have added this information in the text. Line 205.

Results section:

You should give results on several traits such as: means of total and alive born piglets, the farrowing duration, the within-litter weight CV, etc, so that we can better analyze the results. Give results also regarding the experimental piglet’s performance, mortality rate, growth rate, weaning weight.

The number of piglets born from the sows used in the study was 15.2, of which 13.3 were live-born piglets and 1.9 were stillborn and mummified piglets. However, the farrowing duration was not controlled. Growth rate and weight at 10 days was 114.3 ± 63.7g and 1975.5 ± 693.1g respectively. Piglets were not weighed individually at weaning, but litters were weighed. Mean weight was 5.43 ± 1.40kg at 21.00 ± 2.33 days.

I don’t know the potential effects of colostrum on all the various studied parameters however I consider that for some of them, the amount of ingested colostrum could largely influence their values at 72h. For example, the Ig’s levels at 72h are highly influenced by colostrum intake and quality (composition) as colostrum is the only Ig’s source. In a “normal” situation IUGR piglets ingest much less colostrum (and probably with lower Ig levels) than piglets heavier than 1500g, however in your study no differences were observed. Additionally, besides Ig’s, colostrum is rich in maternal cells (mainly lymphocytes and epithelial cells) that can be absorbed by their offspring (Le Jan et al., 1995), so the colostrum intake could also influence lymphocytes levels.

Le Jan, C., Le Dividich, J., Chevaleyre, Hulin, J.C., 1995. Devenir des cellules colostrales chezle porc nouveau-né. Journées de la Recherche Porcine 27, 91–96.

Thank you for your information, we did indeed find no significant differences in immunoglobulin levels.

Reviewer 4 Report

Comments and Suggestions for Authors

Comments to the Authors of manuscript number: animals-2657594 entitled “Comparison of Haematological and Biochemical Profiles be-tween Intrauterine Growth Restriction and Normal Piglets at 72 Hours Postpartum”.

In recent years, the pig industry's genetic selection for increased prolificacy has resulted in diverse litters, an upsurge in low-birth-weight piglets, and a higher occurrence of intrauterine growth restriction (IUGR), adversely affecting piglet survival. This study aimed to investigate the early-life hematological and biochemical profiles of IUGR piglets compared to their normal counterparts. The findings, obtained 72 hours after birth, demonstrated notable differences in blood parameters, reflecting immaturity in IUGR piglets. These variations, including changes in red blood cells, reticulocytes, lymphocyte counts, insulin levels, and oxidative stress, suggest possible vulnerabilities to future health issues, such as metabolic syndrome, making IUGR piglets more susceptible to illness.

1. Introduction: The text contains redundant information. The text lacks clarity and proper organization. It jumps between different aspects of IUGR without a clear flow, making it challenging to follow. The text contains grammatical issues, such as inconsistent verb tenses, which can affect its readability and comprehension.

There is no clear hypothesis.

2. L 120- it should be clearly presented how many piglets were taken from one sow including that control and with IUGR.

3. L 170 -kit number and minimal detection should be given

4. L 172- the same as above

5. L 177 – all kits number should be listed and the minimal detection given by the producer

6. discussion: The text contains complex and lengthy sentences, which can make it challenging for readers to follow the arguments effectively. The discussion repeats certain ideas and findings. The discussion would benefit from a more explicit interpretation of the study's findings. It should connect the observed differences in various parameters to their potential implications for piglet health and development.

7. it is a very interesting study.

Author Response

Dear Reviewer 4. 

Thnak for your suggestions to improve our work. Here are my responses to your comments.

In recent years, the pig industry's genetic selection for increased prolificacy has resulted in diverse litters, an upsurge in low-birth-weight piglets, and a higher occurrence of intrauterine growth restriction (IUGR), adversely affecting piglet survival. This study aimed to investigate the early-life hematological and biochemical profiles of IUGR piglets compared to their normal counterparts. The findings, obtained 72 hours after birth, demonstrated notable differences in blood parameters, reflecting immaturity in IUGR piglets. These variations, including changes in red blood cells, reticulocytes, lymphocyte counts, insulin levels, and oxidative stress, suggest possible vulnerabilities to future health issues, such as metabolic syndrome, making IUGR piglets more susceptible to illness.

Introduction: The text contains redundant information. The text lacks clarity and proper organization. It jumps between different aspects of IUGR without a clear flow, making it challenging to follow. The text contains grammatical issues, such as inconsistent verb tenses, which can affect its readability and comprehension.

Response: Thank you for your comment, the introduction has been improved for better comprehension.

There is no clear hypothesis.

Thanks for your clarification, we have added another paragraph to make the information clearer. Line 104-106. Piglets with IUGR characteristics may exhibit differences in their haematological and biochemical parameters at 72 hours that may affect their growth and development compared to piglets born with normal weight.

L 120- it should be clearly presented how many piglets were taken from one sow including that control and with IUGR.

Response: This study is part of a larger project. Initially the piglets of both groups were selected from the same litter. However, at 72h mortality was higher in IUGR piglets, which made the number of sows greater than 20. In any case, all piglets coincide on the day of birth and no piglet from the control group had IUGR characteristics.

L 170 -kit number and minimal detection should be given

L 172- the same as above

L 177 – all kits number should be listed and the minimal detection given by the producer

The references and ranges of the tests have been incorporated into the text. Line 194-203.

discussion: The text contains complex and lengthy sentences, which can make it challenging for readers to follow the arguments effectively. The discussion repeats certain ideas and findings. The discussion would benefit from a more explicit interpretation of the study's findings. It should connect the observed differences in various parameters to their potential implications for piglet health and development.

it is a very interesting study.

Response: Thank you for your comment, the discussion has been improved for better comprehension.

Round 2

Reviewer 2 Report

Comments and Suggestions for Authors

The Authors have considered all the Reviewer’s comments and corrected article according to the suggestions.

Reviewer 3 Report

Comments and Suggestions for Authors

In fact you still not known if colostrum intake could also influence the studied parameters. You "assume" that colostrum intake was similar but you don't know it for real. In future experiments please consider piglet weighing at 24h.